# Cardiomyopathies: Temporal Review and Genetic Determination

**DOI:** 10.3390/biomedicines13102470

**Published:** 2025-10-10

**Authors:** Gaetano Thiene, Stefania Rizzo, Cristina Basso

**Affiliations:** Department of Cardiac, Thoracic, Vascular Sciences and Public Health, University of Padua, 35121 Padova, Italy; s.rizzo@unipd.it (S.R.); cristina.basso@unipd.it (C.B.)

**Keywords:** cardiomyopathies, genetics, heart transplant, pathology, sudden death

## Abstract

Cardiomyopathies are a heterogeneous group of diseases of the myocardium associated with dysfunction, with or without a structural substrate. They are frequently genetically determined. The dysfunction may be mechanical, both of the systole and diastole, or electrical, including arrhythmias or conduction disorders. Originally, only dilated, hypertrophic, restrictive–obliterative and arrhythmogenic dysfunctions were considered cardiomyopathies. Nowadays, since dysfunction can also be electric, disorders affected by electrical dysfunction without a structural substrate can be regarded as cardiomyopathies as well. This is the case of channellopathies and ryanodine receptors. This paper is a review of the history of cardiomyopathies, including the issues of their classification and nomination, genetic background and gene therapy.

## 1. Introduction

In 1980, John F. Goodwin (1918–2001) (Figure 1), Professor of Cardiology at the Westminster Hospital in London and chair of a committee of the WHO, introduced the term cardiomyopathy to describe heart muscle disease of unknown cause and specific heart muscle disease in which the myocardium is associated with multiorgan involvement.

Goodwin’s paper was published in the *British Heart Journal* with the title “Report of the WHO/ISFC task force on the definition and classification of cardiomyopathies” [2]. Disorders of the myocardium due to systemic or pulmonary hypertension, coronary artery, cardiac valves and congenital heart diseases were ruled out.

## 2. Historical Overview

Cardiomyopathies were initially classified as dilated (Figure 2a), hypertrophic (Figure 2b) and restrictive–obliterative (Figure 2c), the latter meaning obliteration of ventricular cavities by eosinophilic endomyocardial inflammation with thrombosis (Loeffler heart disease) (Figure 3). Eosinophilia may be due to eosinophil leukemia or due to intestinal infection by helminths, known in Africa by the name of Davies disease.

Concerning dilated cardiomyopathy, ventricular dilatation (Figure 2a and Figure 4a) can be due to poor contractility caused by myocytolysis (Figure 4b) or previous viral myocarditis with scarring from necrosis (Figure 5).

As for hypertrophic cardiomyopathy (Figure 2b), it is characterized by asymmetric hypertrophy of the left ventricle, with myocardial disarray at histology (Figure 6), a bizarre tridimensional arrangement of cardiomyocytes initially interpreted as cardiac hamartoma [1].

## 3. Pathophysiological Classification of Specific Cardiomyopathies

Among specific heart muscle diseases, a series of morbid entities were included [2]:Myocarditis [5].Metabolic diseases (endocrine, storage, amyloid).General system diseases (connective tissue disorders, sarcoidosis, leukemia).Heredofamilial (muscular dystrophies, neuromuscular disorders).Sensitivities and toxic reactions (sulphonamides, alcohol, anthracyclines, irradiation).

Regarding inflammatory myocardial diseases, Fiedler myocarditis [6] (Figure 7) was named “isolated”, because only the heart was affected, and “interstitial” because the myocardial interstitium was infiltrated by inflammatory cells, unlike diphtheric myocarditis, in which the damage was only parenchymal. In Fiedler myocarditis, the lymphocytic inflammatory infiltrate (Figure 7a) was most probably due to a viral infection; however, at that time, molecular diagnosis for viral detection was not yet available. A giant cell pattern was also observed in Fiedler myocarditis (Figure 7b), probably ascribable to immune reaction. Sarcoid myocarditis, with non-caseous granuloma, is also most probably immune-related in terms of pathophysiology (Figure 8). It may involve the lymph nodes and other organs like the lungs.

## 4. Storage- and Interstitial-Specific Cardiomyopathies

Intracellular storage occurs in hemochromatosis (Figure 9), glycogenosis (Figure 10) and Fabry’s disease. Amyloidosis deposits occur (Figure 11) in the interstitium and in the small vessel wall.

## 5. Chronology of WHO Classification

After the WHO classification of 1980 [7], another two cardiomyopathies were discovered: arrhythmogenic right ventricular [8] (Figure 12) and, more recently, arrhythmogenic left ventricular (Figure 13). They are characterized by fibrofatty replacement of the myocardium in the subepicardium and are at high risk of arrhythmic sudden death.

The second novel type of cardiomyopathy is restrictive, not obliterative, characterized by stiff diastole with poor ventricular relaxation and by atrial dilatation in the absence of ventricular hypertrophy. The heart is small, and the diastolic ventricular filling is hindered by severe congestive heart failure. When compared with dilated cardiomyopathy, restrictive cardiomyopathy represents the paradox of a small heart requiring transplantation (Figure 14). The histology of the ventricular myocardium shows myocardial disarray, like in hypertrophic cardiomyopathy (Figure 15b). Myocardial disarray accounts for impairment of diastolic ventricular filling. This is the reason why both cardiomyopathies are characterized by atrial fibrillation. Restrictive cardiomyopathy is nowadays jokingly called “hypertrophic cardiomyopathy without hypertrophy”.

The discovery of arrhythmogenic cardiomyopathy [8] and restrictive cardiomyopathy [1] required a review of the WHO classification. In June 1995, the WHO committee met in Genève, chaired by Paul Richardson. A new definition and classification was advanced [9].

## 6. Storytelling of Definition and Classification of Cardiomyopathies

Table 1 and Table 2 compare the definitions and classifications of 1980 vs. 1996. As for the definition of cardiomyopathy, it was changed from “heart muscle disease of unknown cause” to “disease of the myocardium associated with cardiac dysfunction”. And the definition of specific heart disease was changed in specific cardiomyopathy, so that the term cardiomyopathy was employed for any heart muscle disease (Table 1).

New entities (arrhythmogenic and restrictive non obliterative) were added to the cardiomyopathy classification (Table 2). The new classification was published in *Circulation* in 1996 [9].

Some cardiomyopathies remained unclassified (Table 3).

Endocardial fibroelastosis (=dilated cardiomyopathy in children) (Figure 16) appears among unclassified cardiomyopathies both in 1980 and 1995 (Table 3).

More recently, in 1997, the aetiology of endocardial fibroelastosis was determined to be an infection in the uterus due to mumps virus of the myocardium thanks to molecular investigation [10,11].

Histiocytoid cardiomyopathy was definitively interpreted as a tumour of Purkinje cells (Purkinjoma) (Figure 17) [12].

In 1996, a non-compacted left ventricular myocardium (Figure 18) was still considered an unclassified cardiomyopathy. Lack of ventricular wall compaction is an embryological defect, so the disease should be considered a congenital heart disease [13].

Nowadays, Fiedler’s myocarditis [6] is considered a cardiomyopathy, as any other viral or immunological myocarditis.

In 2006, the definition and classification of cardiomyopathies also attracted the interest of the American Heart Association [15]. Table 4 reports the 2006 AHA definition of cardiomyopathies. The goals of the 2006 AHA classification were as follows:Electrical heart dysfunction is a myocardial dysfunction, so channelopathies are cardiomyopathies.When myocardial dysfunction is the consequence of other cardiovascular diseases (valve, hypertension, congenital, coronary artery), it is excluded from the classification of cardiomyopathies.Myocarditis is a cardiomyopathy.

Cardiomyopathies are a major cause of severe heart failure requiring cardiac transplantation (Table 5). In the experience (1985–2015) of the University of Padua, cardiomyopathies accounted for 51.4% of heart recipients (Table 5) [3].

Among cases of sudden death in 650 young people, 31.3% of patients died because they were affected by cardiomyopathies: hypertrophic cardiomyopathy, arrhythmogenic cardiomyopathy and myocarditis (Table 6) [3].

## 7. Genetic Background of Cardiomyopathies

Cardiomyopathies are usually genetically determined. Hippocrates stated that diseases may be handed down from parents to offspring from the very moment of conception. Table 7 reports the classification of inherited cardiomyopathies caused by defective genes and wrong-coded proteins: cytoskeleton, sarcomere, desmosome, and ion channels.

Dilated cardiomyopathy is mostly related to gene defects of cytoskeleton proteins of the nuclear and cell membranes. Missense mutation of lamin A/C and truncation of titin give origin to genetically determined cardiomyopathies, the former also accounting for AV conduction disturbances.Hypertrophic and restrictive cardiomyopathy are both due to mutations of genes coding sarcomere proteins [16,17].Arrhythmogenic cardiomyopathy is the consequence of mutations of genes coding desmosome proteins (Figure 19) [18].Long QT (Figure 20a), short QT (Figure 20b), Brugada syndrome with non-ischemic ST elevation (Figure 20c), polymorphic catecholaminergic ventricular tachycardia (Figure 20d) and Lenegre disease (Figure 21) are ion channel and ryanodine receptor cardiomyopathies.

Also, complete AV block may be inherited, with the name of Lenegre disease (Figure 21).

Thus, mutations of sodium “channel I5” account for long QT, Brugada syndrome and Lenegre disease. The latter is an example of a genetically determined cardiomyopathy of the conduction tissue (Figure 21).

They should be considered cardiomyopathies, regardless of the type of cardiac dysfunction: contractile, electric, intercellular junction, electromechanical association or electric stimulus conduction.

Refs. [17,18,20,21,22,23,24,25,26,27,28] deal with the discoveries of the genetic background of cardiomyopathies. A proposal was advanced of a genetic classification for inherited cardiomyopathies.

## 8. Is It Time for Gene Therapy?

Genetically determined cardiomyopathies are caused by mutations or deletions of DNA.

Recent studies suggested that gene therapy may be a potent molecular option for the treatment of genetically determined cardiomyopathies, like for hypertrophic cardiomyopathy due to myosin mutations and for catecholaminergic polymorphic ventricular tachycardia. Gene therapy is based upon the introduction of genetic material into cardiomyocytes.

There are several strategies involving DNA and RNA messengers.

## 9. DNA Action Strategies for Genetic Therapy [29]

Gene replacement by viral vector, with the delivery of a healthy gene copy, introduced into the cardiomyocyte by a viral vector. DNA adenovirus is the most used. However, adenovirus is a frequent etiology of viral myocarditis, so using this vector may result in a huge inflammatory response, with a risk of iatrogenic death.Modification of signal pathways.Inactivation of mutant gene by cleavage of the endogenous DNA to prevent its expression.Mutation repair by restoring healthy genes after cleavage.Oligonucleotides are short DNA or RNA molecules that can be used to perturb gene expression in target cells. They may be expressed by viral vectors or chemically synthesized and delivered systemically.Modified RNA messengers may be synthesized by in vitro transcription using modified nucleotides, designed to allow them to enter inside the cardiomyocytes without immune reaction and rapid translation into protein.

## 10. Final Reflections

Thanks to the discovery of DNA by James Watson and Francis Crick (Figure 22) [30] and the invention of polymerase chain reaction by Kary Banks Mullis (Figure 23) [31], molecular diagnosis of inherited heart disease is feasible even at autopsy and can be specific. Gene therapy is a life-saving novelty that can prevent the transmission of inherited disease to offspring.

Just like vaccinations led to the disappearance of lethal infective diseases like smallpox, with gene therapy, diseases transmissible to the offspring may also disappear.

Indeed, the dream is the disappearance of every genetic disease in the future, including genetically determined cardiomyopathies (Figure 24), both with and without (Figure 25 and Figure 26) structural defects, both of which pose an equal risk of sudden death.

In 1894, Karl von Rokitansky and Rudolph Virchow attended a meeting on “Morgagni and the Anatomic Concept” in Rome. Virchow raised the following questions: “…Any anatomic modification is material, but is any material modification anatomic? Why not molecular? Can a profound molecular modification occur in the setting of an apparently normal structure? These modifications belong more to physiology than to anatomy, they are functional-dynamic… the method of investigation will never be morphological”.

Refs. [34,35,36,37,38,39] provide further contributions on gene therapy for inherited and genetically determined cardiomyopathies.

## Figures and Tables

**Figure 1 biomedicines-13-02470-f001:**
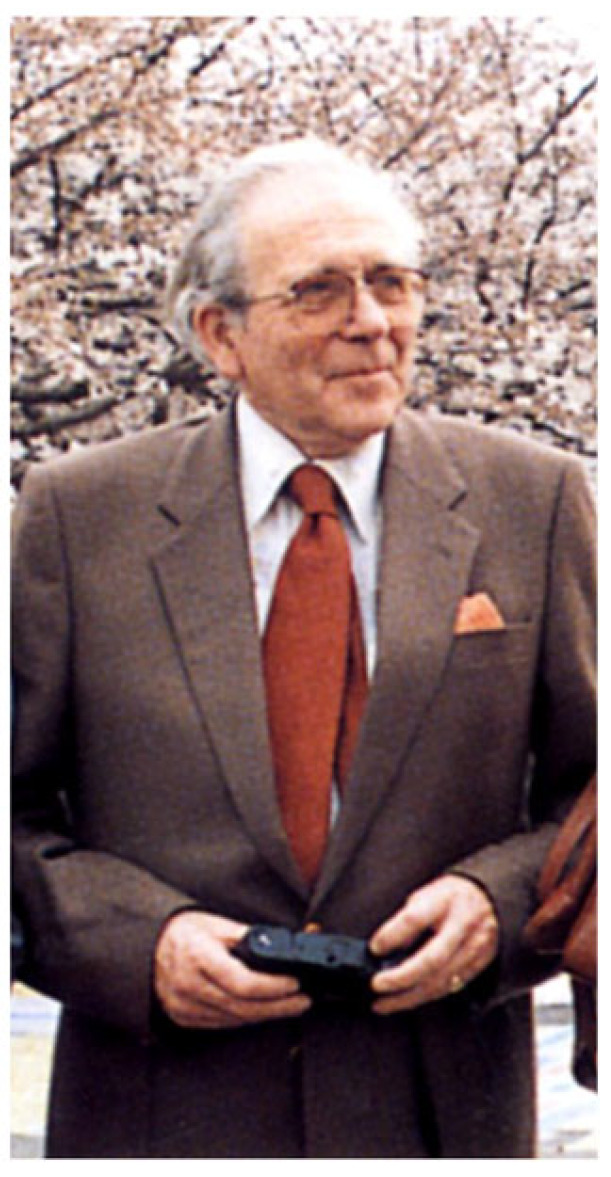
John F. Goodwin, the father of the classification, definition and nomination of cardiomyopathies. He used the term cardiomyopathy for heart muscle diseases of unknown cause and specific heart muscle diseases associated with morbidities of other systems. Modified from [1].

**Figure 2 biomedicines-13-02470-f002:**
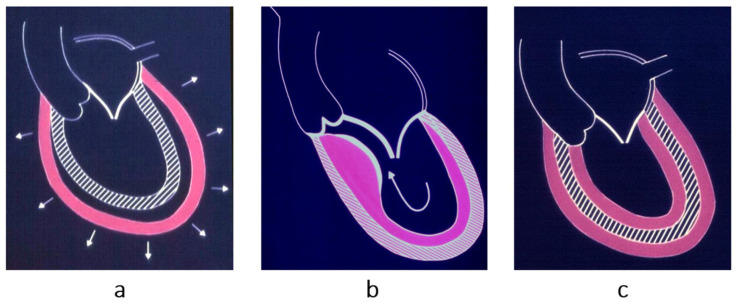
The original classification of cardiomyopathies by John Goodwin [2] was as follows: (**a**) dilated, when the ventricular cavities appear increased in size; The arrows indicate the progressive enlargement of the ventricles in dilated cardiomyopathy; (**b**) hypertrophic, when the free wall or septum are thickened in an asymmetric way; (**c**) restrictive, when the ventricular cavities are reduced in size or even obliterated, with a thickened endocardium and thrombus. The problem is the reduced ventricular cavity, not the stiffness of the ventricular wall, hindering diastolic relaxation [1]. From [1] with permission.

**Figure 3 biomedicines-13-02470-f003:**
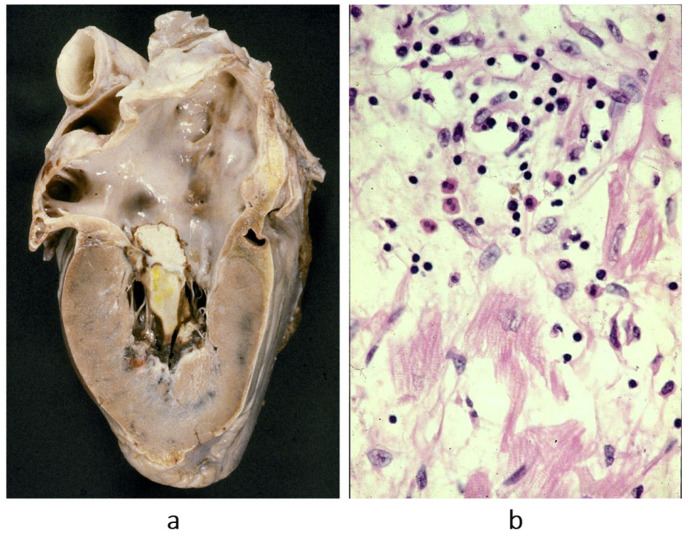
Obliterative cardiomyopathy (**a**) due to endomyocardial eosinophilic inflammation (haematoxylin–eosin stain) (**b**), with thrombosis, known by the name of Loeffer cardiomyopathy in the setting of eosinophil leukemia or by the name of Davies disease when due to intestinal helminth infection with allergic eosinophil inflammation. From [3] with permission.

**Figure 4 biomedicines-13-02470-f004:**
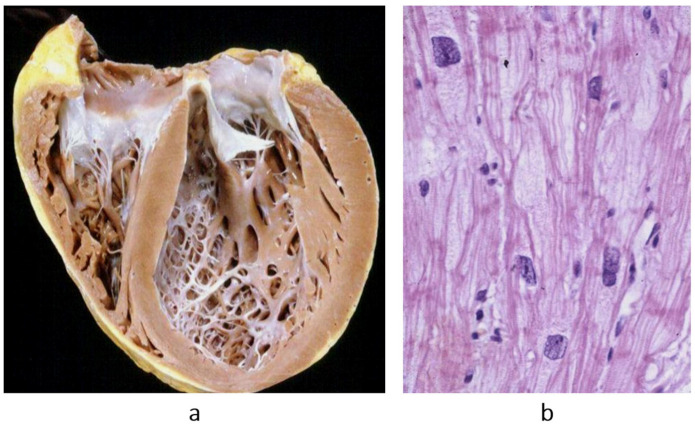
Example of a heart with dilated cardiomyopathy, gross view (**a**), and myocytolysis of cardiomyocytes on histology (**b**), haematoxylin–eosin stain. From [3] with permission.

**Figure 5 biomedicines-13-02470-f005:**
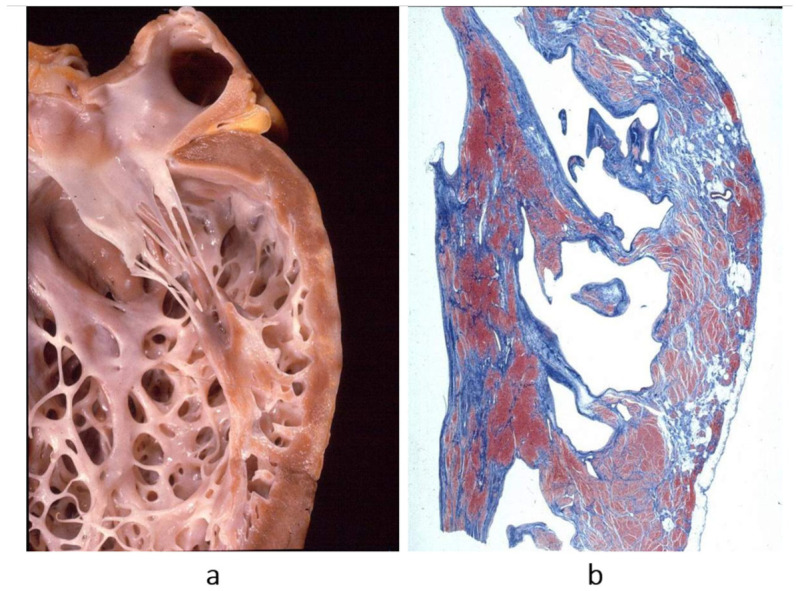
Another type of dilated cardiomyopathy (**a**) with severe, non-ischemic myocardial fibrosis (**b**). The patient previously had viral myocarditis, which accounted for myocyte necrosis and scarring. Azan Mallory stain. From [3] with permission.

**Figure 6 biomedicines-13-02470-f006:**
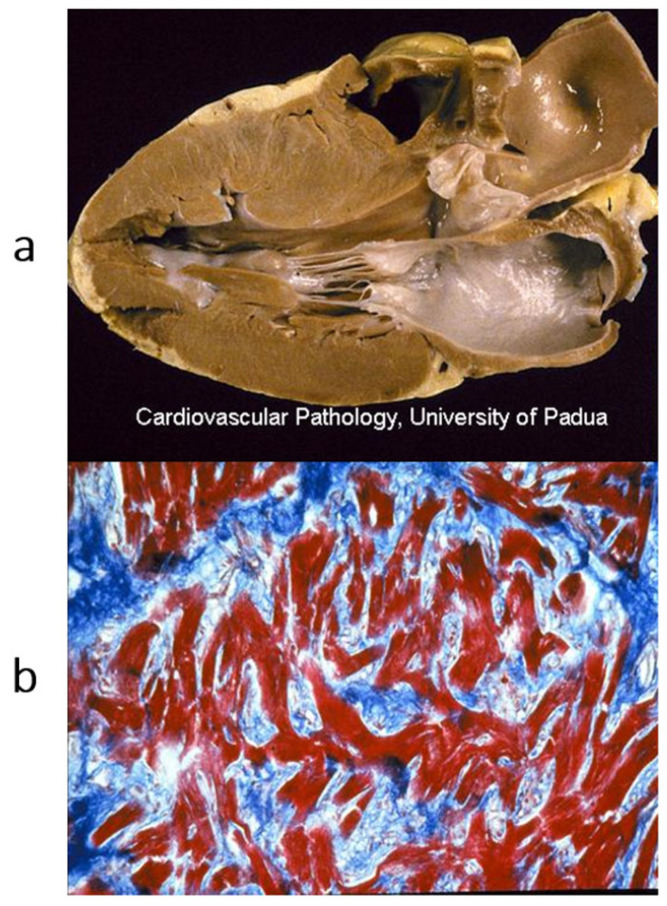
(**a**) Heart specimen with hypertrophic cardiomyopathy due to asymmetric ventricular septum thickness. Hypertrophy, long-axis view. (**b**) Disarray of cardiomyocytes on histology. Originally, the disarray was interpreted as a benign tumour (heart hamartoma). Azan Mallory stain. From [4] with permission.

**Figure 7 biomedicines-13-02470-f007:**
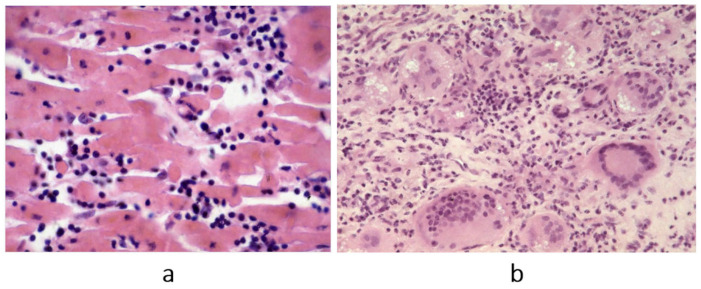
Fiedler’s myocarditis on microscopy. A review of the original slides demonstrates two histology patterns: (**a**) lymphocytic inflammatory infiltrates, most probably due to a virus (molecular analysis was not available in that time); (**b**) immunological inflammation of giant cells. Haematoxylin–eosin stain. From [5] with permission.

**Figure 8 biomedicines-13-02470-f008:**
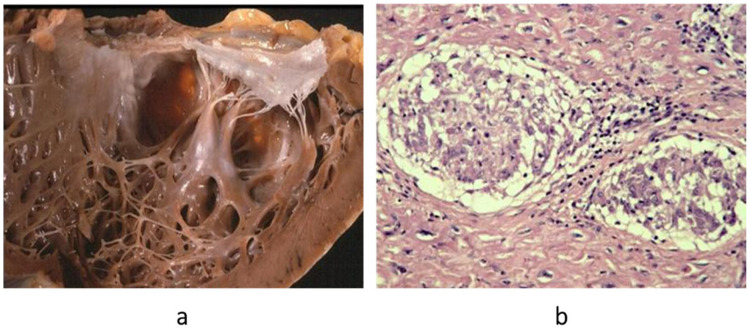
Non-caseous granuloma, typical of sarcoid myocarditis. (**a**) Aneurysm of the left ventricle; (**b**) giant cells in the setting of non-caseous granuloma. Haematoxylin–eosin stain. From [3] with permission.

**Figure 9 biomedicines-13-02470-f009:**
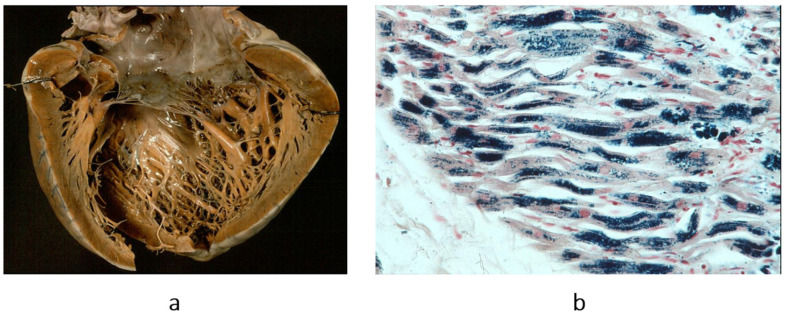
Hemochromatosis cardiomyopathy due to multiple blood infusions. (**a**) The left ventricle shows a brown colour; (**b**) intracellular iron storage. Iron histochemical staining (personal archive).

**Figure 10 biomedicines-13-02470-f010:**
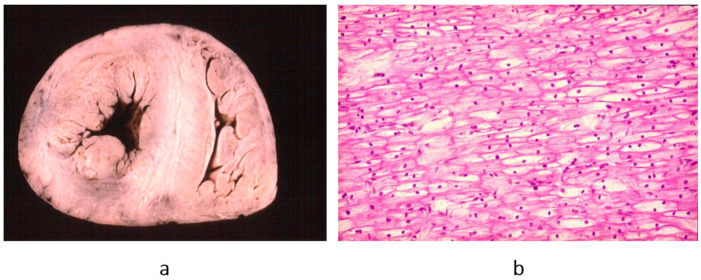
Glycogenosis: (**a**) concentric ventricular hypertrophy with a pale myocardium, cut in short-axis view; (**b**) cardiomyocytes appear empty because glycogen storage was mostly removed by technical procedures (personal archive).

**Figure 11 biomedicines-13-02470-f011:**
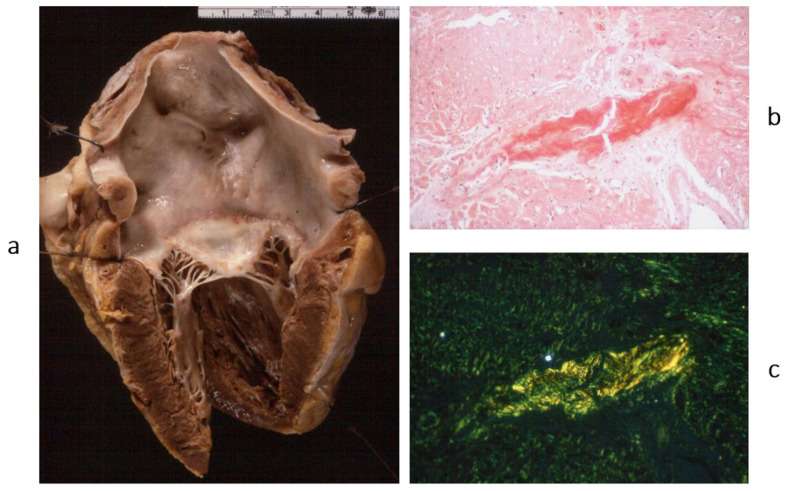
Amyloid deposits in the interstitium of the ventricular myocardium: (**a**) The left cardiac chamber appears dilated and stiff, particularly the left atrium. The patient suffered from atrial fibrillation. (**b**) The amyloid deposits were extracellular within the interstitium and stained with Congo Red and (**c**) Thioflavin-T stain. From [3] with permission.

**Figure 12 biomedicines-13-02470-f012:**
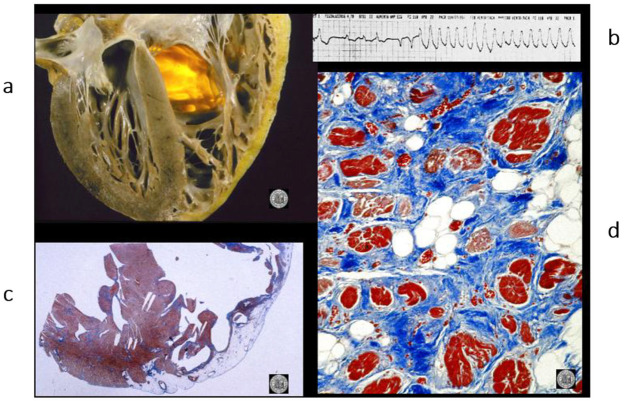
A case of an athlete who died suddenly from right ventricular arrhythmogenic cardiomyopathy during tennis play. (**a**) Four-chamber cut. Gross appearance of the right ventricle with transmural fibrofatty replacement and translucent free wall. (**b**) ECG recording ventricular tachyarrhythmia. (**c**) Compare the histology of the left and right ventricles. The former has an intact myocardium; the latter has transmural fibrofatty tissue. Also, the ventricular septum is intact (Azan Mallory stain). (**d**) Histology of the right ventricle at higher magnification, with fibrofatty replacement (Azan Mallory stain). From [4] with permission.

**Figure 13 biomedicines-13-02470-f013:**
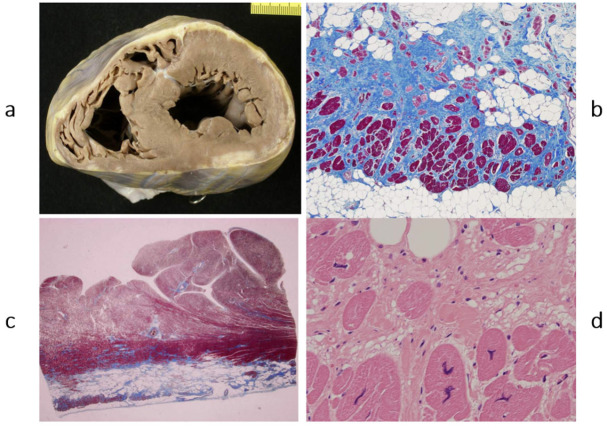
Sudden death from arrhythmogenic cardiomyopathy of the left ventricle. It occurred in a 27-year-old hockey player. (**a**,**b**) Spots of fibrofatty replacement of the myocardium of the subepicardium of the left ventricle, gross and histologic views. (**c**,**d**) Other histologic slides at high magnification, showing fibrofatty replacement. Azan Mallory stain (personal archive).

**Figure 14 biomedicines-13-02470-f014:**
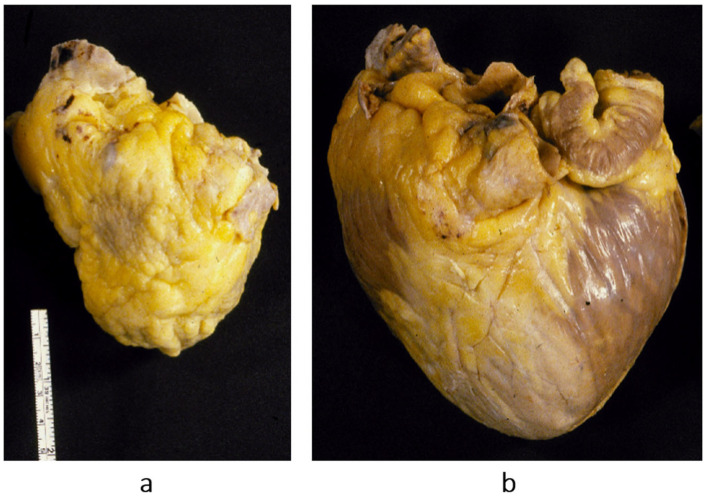
The paradox of a small heart with restrictive cardiomyopathy (**a**), compared with dilated cardiomyopathy (**b**). Both were patients with severe congestive heart failure requiring transplant. From [1] with permission.

**Figure 15 biomedicines-13-02470-f015:**
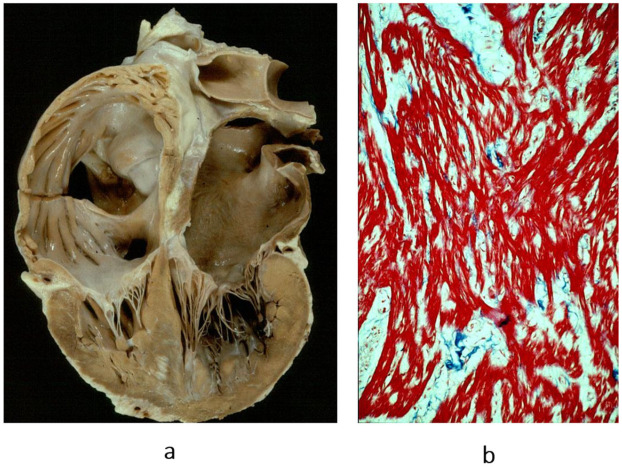
Restrictive cardiomyopathy (not obliterative). (**a**) Gross view of four-chamber section showing dilated atria. Note that the ventricular chambers do not exhibit obliteration. The hindered ventricular diastole-like appendages of the ventricles are due to stiff ventricular walls. (**b**) Histology show myocardial disarray, similar to that in hypertrophic cardiomyopathy. Azan Mallory stain.

**Figure 16 biomedicines-13-02470-f016:**
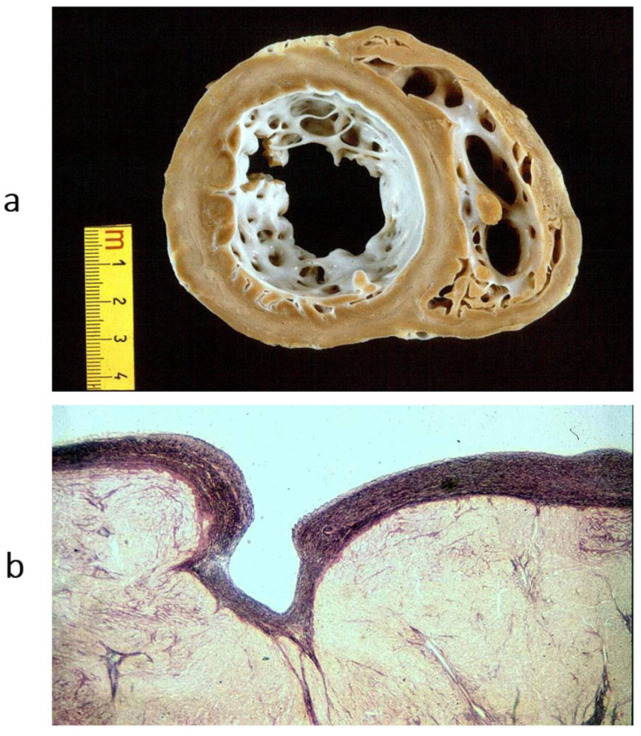
Endocardial fibroelastosis, considered for years an unclassified cardiomyopathy. The trouble was whether to consider it a congenital heart disease (because it is due to prenatal infection) or an early onset of cardiomyopathy. Eventually, it was discovered that it is the consequence of an infection with mumps virus, transmitted at the time of conception by the parents. (**a**) A thick, white endocardium is visible on the cross-section of the left ventricles. (**b**) Close-up of the endocardium on histology; it appears thickened by proliferation of elastic fibres. Weigert–Van Gieson stain. From [3] with permission.

**Figure 17 biomedicines-13-02470-f017:**
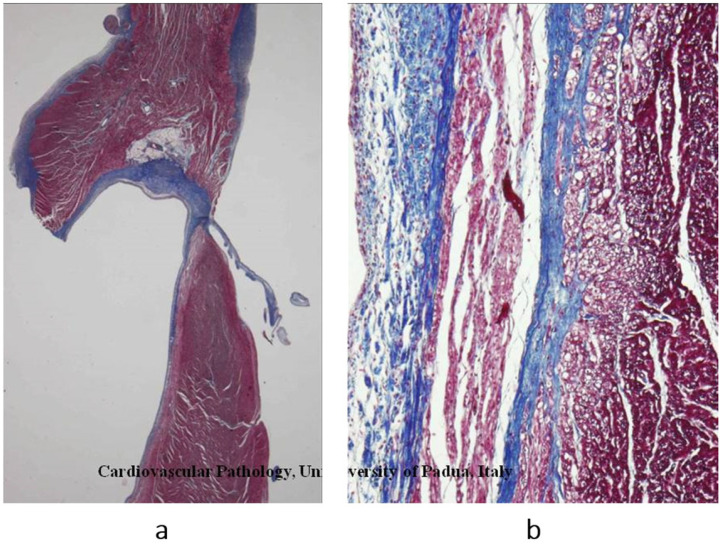
Proliferation of Purkinje cells: cardiomyopathy, congenital heart disease or multifocal tumour of conduction tissue? The histology clearly shows an exuberant number of Purkinje cells, in keeping with a neoplastic phenomenon. (**a**) Histologic section of the atrioventricular junction and the AV conduction system; (**b**) proliferation of Purkinje cells. Azan Mallory stain (personal archive).

**Figure 18 biomedicines-13-02470-f018:**
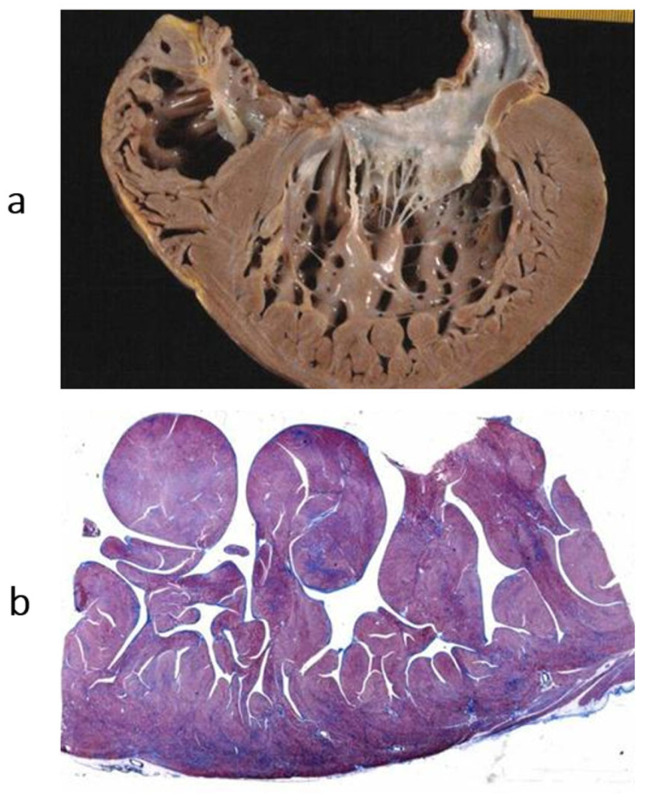
Non-compacted left ventricle. (**a**) Gross view of the left ventricular cavity, seen in the four-chamber section. The cavity is full of coarse trabeculae. On echo, the diagnosis that was put forward was dilated cardiomyopathy with mural thrombi. The patient underwent successful cardiac transplantation. (**b**) The histology of the trabeculae shows the endocardium almost touching the epicardium. The disease is a congenital heart disease because of embryological failure that occurred due to compacting myocardial trabeculae of the left ventricle. Haematoxylin–eosin stain. From [14] with permission.

**Figure 19 biomedicines-13-02470-f019:**
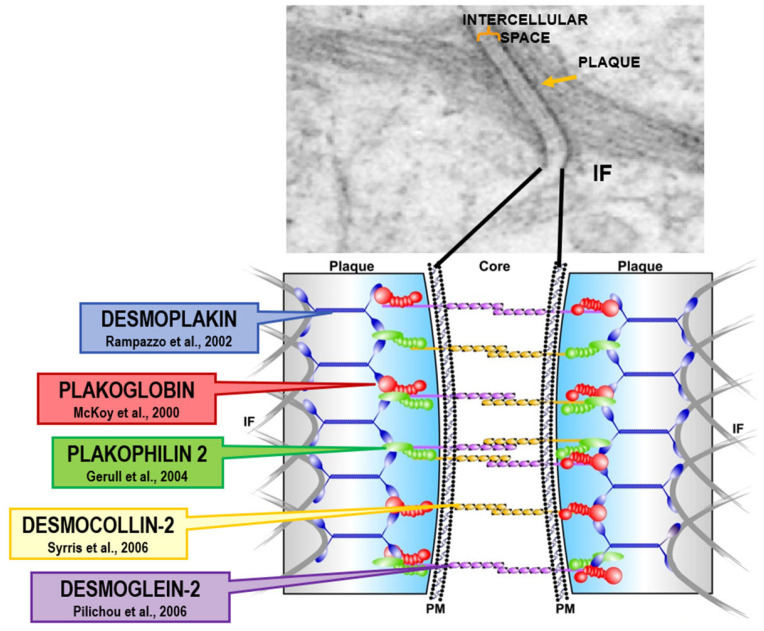
Arrhythmogenic cardiomyopathy is a genetically determined cardiomyopathy caused by missense dominant mutations of genes coding desmosomal proteins of the intercellular junction. Recessive arrhythmogenic cardiomyopathy of the Naxos isle is due to deletion of Plakoglobin gene. From [19] with permission.

**Figure 20 biomedicines-13-02470-f020:**
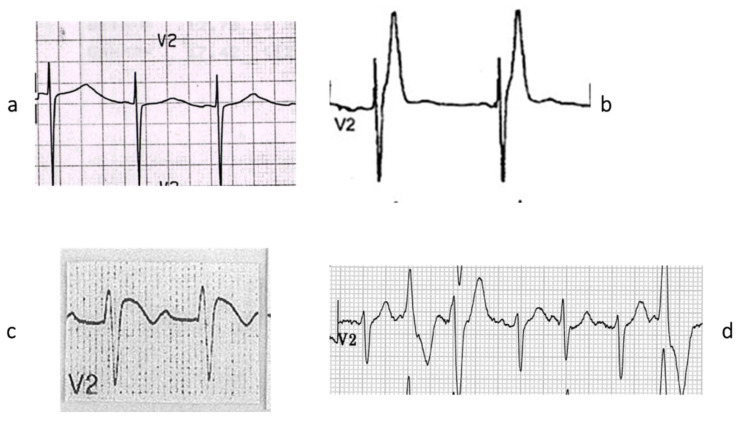
There are genetically determined cardiomyopathies without structural substrate and electrical dysfunction. This is the case of long QT (**a**) and short QT (**b**) interval and non-ischemic ST segment elevation in Brugada syndrome (**c**), called channelopathies of sodium, potassium, and polymorphic ventricular tachycardia caused by calcium ryanodinic receptor due to their electro-anatomic mechanical association (**d**) (personal archive).

**Figure 21 biomedicines-13-02470-f021:**
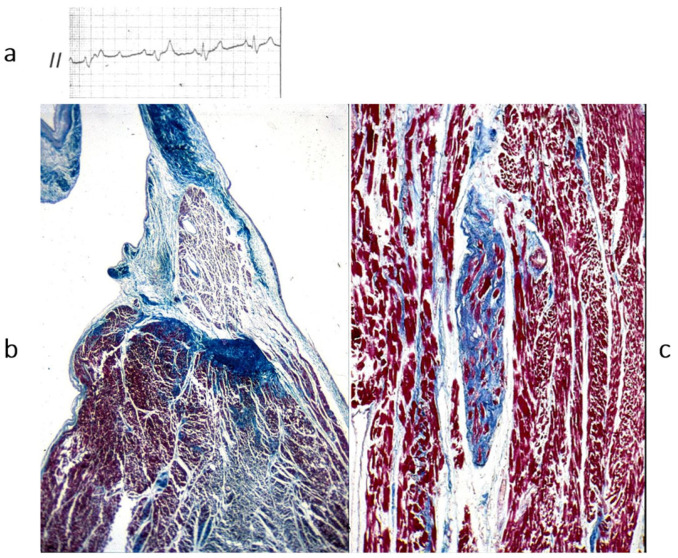
Also, AV block may be to a sodium channelopathy. (**a**) AV block on ECG; (**b**) fibrotic disruption of bundle of His bifurcation; (**c**) right bundle branch is replaced by fibrosis. Azan Mallory stain. The inherited AV block of Lenegre should be considered a cardiomyopathy of the conduction system. From [4] with permission.

**Figure 22 biomedicines-13-02470-f022:**
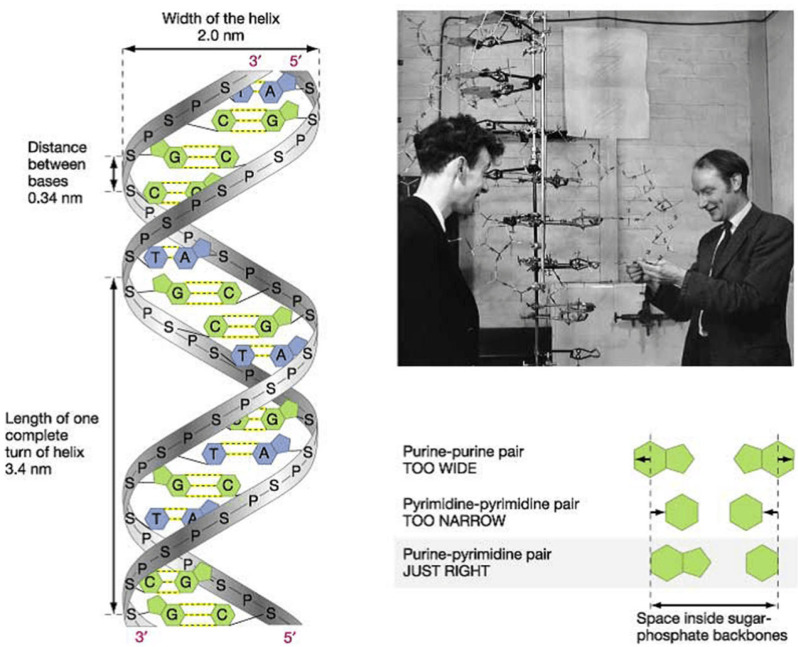
In 1953, James Watson and Francis Crick co-discovered the DNA double helix (public domain).

**Figure 23 biomedicines-13-02470-f023:**
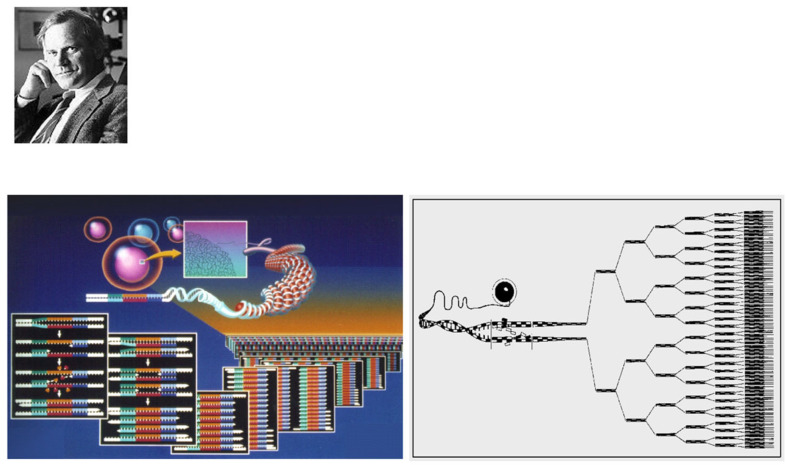
In 1985, Kary Banks Mullis invented the polymerase chain reaction (PCR), a revolution for investigating genetic diseases. From [14] with permission.

**Figure 24 biomedicines-13-02470-f024:**
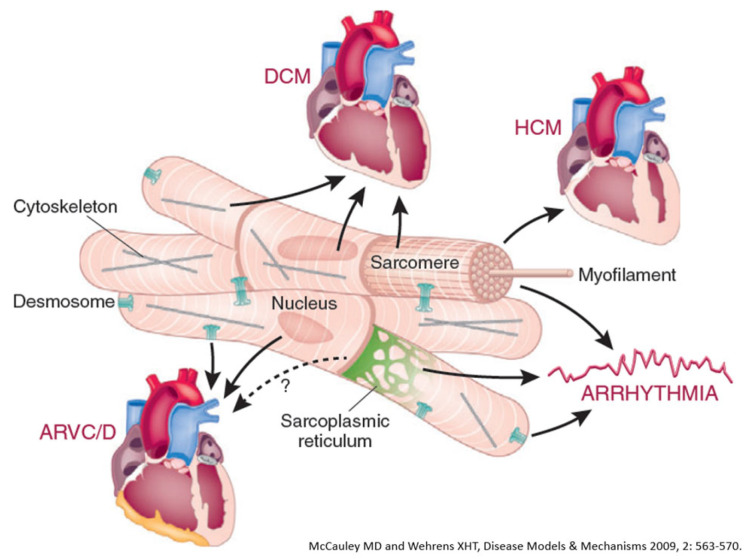
Inherited cardiomyopathies with structural abnormalities: dilated cardiomyopathy (DCM), hypertrophic cardiomyopathy (HCM) and arrhythmogenic right ventricular cardiomyopathy (ARVCM). From [32] with permission.

**Figure 25 biomedicines-13-02470-f025:**
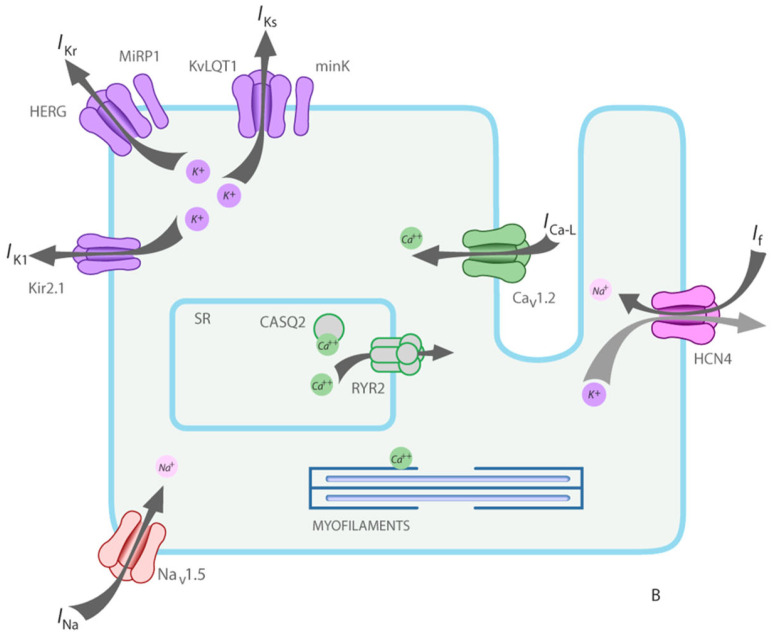
Inherited cardiomyopathies without structural abnormalities due to gene mutation of ion channels and ryanodine receptor. From [33] with permission.

**Figure 26 biomedicines-13-02470-f026:**
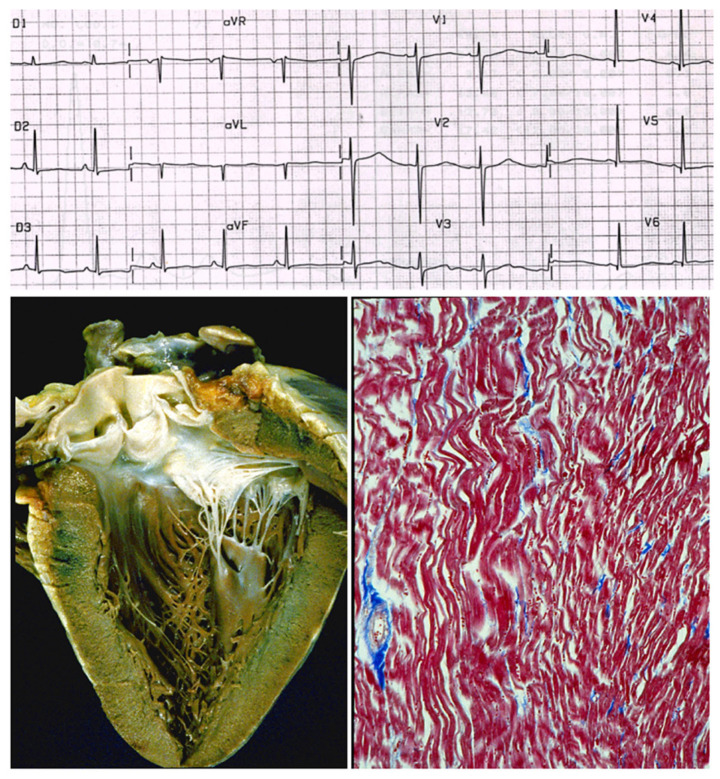
A case of sudden death in a young person due to long QT without structural defects. The ECG shows the long QT, the heart specimen does not exhibit structural defects, and the histology of the myocardium is normal. Azan Mallory stain. Is it a cardiomyopathy? Is long QT a congenital heart disease? From [4] with permission.

**Table 1 biomedicines-13-02470-t001:** **Definitions:** 1980. vs. 1996.

1980	1996
Cardiomyopathy, definition	Cardiomyopathy, definition
Heart muscle disease of unknown cause	Disease of the myocardium associated with cardiac dysfunction
Specific heart muscle disease, definition	Specific cardiomyopathy, definition
Heart muscle disease of unknown cause or associated with disorders of other systems	Heart muscle disease associated with specific systemic disorders

**Table 2 biomedicines-13-02470-t002:** Update of the WHO classification of cardiomyopathies: 1980 vs. 1996.

1980 Classification	1996 Classification
Dilated	Dilated
Hypertrophic	Hypertrophic
Obliterative	Restrictive
	Arrhythmogenic right ventricular

**Table 3 biomedicines-13-02470-t003:** WHO unclassified cardiomyopathies.

1980 Unclassified	1996 Unclassified
Endocardial fibroelastosis	Endocardial fibroelastosis
Histiocytoid cardiomyopathy	Non compacted myocardium
Fiedler’s myocarditis	Mitochondrial cardiomyopathy

**Table 4 biomedicines-13-02470-t004:** The 2006 AHA definition of cardiomyopathies.

2006 AHA Definition of Cardiomyopathies	Cardiomyopathies are a heterogeneous group of diseases of the myocardium associated with mechanical and/or electrical dysfunction that usually (but not invariably) exhibit inappropriate ventricular hypertrophy or dilatation and are due to a variety of causes, which are frequently genetic.

**Table 5 biomedicines-13-02470-t005:** Cardiomyopathy in 876 cases of patients who underwent orthotopic cardiac transplantation (1985–2015).

	%
Dilated Cardiomyopathy	38.7
Hypertrophic Cardiomyopathy	3.0
Restrictive Cardiomyopathy	2.6
Arrhythmogenic Cardiomyopathy	4.2
Myocarditis	2.9
**TOTAL**	**51.4**

**Table 6 biomedicines-13-02470-t006:** Cardiomyopathies among 650 cases of sudden cardiac death in the young (1980–2016).

	%
Dilated Cardiomyopathy	0.3
Hypertrophic Cardiomyopathy	9.0
Restrictive Cardiomyopathy	0.0
Arrhythmogenic Cardiomyopathy	10.0
Myocarditis	12.0
**TOTAL**	**31.3**

**Table 7 biomedicines-13-02470-t007:** Genetical determined cardiomyopathies.

Name	Phenotypic Expression
Dilated	=cytoskeleton cardiomyopathy
Hypertrophic and restrictive	=sarcomere cardiomyopathy
Arrhythmogenic RV, Naxos diseases	=desmosomal cardiomyopathy
Long and short QT syndromes, Brugada syndrome, catecholaminergic polymorphic VT, Lenegre disease	=ion channel, cardiomyopathy =ryanodine receptor

## Data Availability

No new data were created or analyzed in this study.

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
