# Peer review of "Cardiomyopathies: Temporal Review and Genetic Determination"

_biomedicines, 2025, doi:10.3390/biomedicines13102470_

Round 1

Reviewer 1 Report

Comments and Suggestions for Authors

The manuscript and its content is really original and well written. Figures are well represented. It is interesting the definition of CMP also for electrical disorders. Moreover it could open the way to a more integrated approach when dealing with heart muscle disease and genetic electrical disorders. 

Author Response

I appreciate your positive evaluation of the manuscript, particularly the agreement to include myocyte disorders with electrical dysfunction within cardiomyopathies.

Reviewer 2 Report

Comments and Suggestions for Authors

The manuscript is devoted to the classification of cardiomyopathies, genetic causes of cardiomyopathies and the prospects of gene therapy. In present form, the manuscript is characterized by limited novelty and requires serious revision.

Major remark

  1. Most of the Introduction Section is devoted to the history of the classification of cardiomyopathies and only a small part of it talks about genetically determined cardiomyopathies. Since Dr. G. Thiene previously published reviews with a detailed description of the history of the classification of cardiomyopathies, a description of the types of cardiomyopathies, their characteristics (McKenna WJ, Maron BJ, Thiene G. Classification, Epidemiology, and Global Burden of Cardiomyopathies. Circ Res. 2017;121(7):722-730. doi: 10.1161/CIRCRESAHA.117.309711; Thiene G, Calore C, De Gaspari M, Basso C. Storytelling of Hypertrophic Cardiomyopathy Discovery. J Cardiovasc Dev Dis. 2024;11(10):300. doi: 10.3390/jcdd11100300.), then it is necessary to shorten this part of the Introduction in the presented manuscript.

At the same time, it is necessary significantly expand the part about genetically determined cardiomyopathies. To develop the topic of the manuscript, authors must provide the names of genes, whose mutations lead to the corresponding types of cardiomyopathy, provide data on the occurrence of mutations and mechanisms of development of different types of cardiomyopathy. For an example, see https://doi.org/10.1016/B978-0-12-405206-2.00006-5

  1. Most of the Figures and Tables in the submitted manuscript have been published in previous articles by the authors of Manuscript and other authors but there is no indication of receiving permission to publish it from the original source. For example:

Figure 2 was previously published in Thiene et al., J Cardiovasc Dev Dis. 2024.

Figures 3, 5, and 12 were published in McKenna WJ, Maron BJ, Thiene G. Classification, Epidemiology, and Global Burden of Cardiomyopathies. Circ Res. 2017;121(7):722-730. doi: 10.1161/CIRCRESAHA.117.309711.

Figure 4 was previously published in Thiene G. Storytelling of Myocardial Biopsy. Biology (Basel). 2025;14(3):306. doi: 10.3390/biology14030306. And in McKenna WJ, Maron BJ, Thiene G. Classification, Epidemiology, and Global Burden of Cardiomyopathies. Circ Res. 2017;121(7):722-730. doi: 10.1161/CIRCRESAHA.117.309711.

Figure 7 was published in Thiene G. Storytelling of Myocarditis. Biomedicines. 2024;12(4):832. doi: 10.3390/biomedicines12040832.

Figure 13 was published in Thiene G. Arrhythmogenic cardiomyopathy: from autopsy to genes and transgenic mice. Cardiovasc Pathol 2012;21:229–39.

Figure 21 and 22 were published in Franz WM, Müller OJ, Katus HA. Cardiomyopathies: from genetics to the prospect of treatment. Lancet. 2001;358(9293):1627-37. doi: 10.1016/S0140-6736(01)06657-0.

Figure 29 was published in Bhuiyan ZA, Al-Shahrani S, Al-Aama J, Wilde AA, Momenah TS. Congenital Long QT Syndrome: An Update and Present Perspective in Saudi Arabia. Front Pediatr. 2013;1:39. doi: 10.3389/fped.2013.00039.

Tables 1 and 2 were published in McKenna WJ, Maron BJ, Thiene G. Classification, Epidemiology, and Global Burden of Cardiomyopathies. Circ Res. 2017;121(7):722-730. doi: 10.1161/CIRCRESAHA.117.309711. etc

  1. High proportion of self-citations: of 14 sources out of 36. This proportion will decrease if the part of the manuscript devoted to genetically determined cardiomyopathies is significantly expanded.
  2. Page 2, paragraph 2. It is necessary to point out that in the classification proposed by John F. Goodwin, obliterative and restrictive cardiomyopathy are one and the same.

A small note - In the Manuscript, the phrase is very common «As far as».

Figure 17 has no meaning.

Figure 21 has no title or description.

Author Response

  • The introduction has been integrated by mentioning genetically determined cardiomyopathies.
  • The introduction was shortened by reducing Fig. 1, leaving only the photo of John F. Goodwin and by quoting relevant previous papers of the first author, removing names and titles of papers.
  • Genetically determined cardiomyopathies and name of wrong coded proteins phenotypic expression has been reported at page ……. in TABLE 7 and Figs. 20, 21.
  • As far as previous published figures, all has been employed with the permission for publication.
  • Five self-citations have been deleted.
  • The phrase as far as has been used the less the possible.
  • 17 has been eliminated.

Reviewer 3 Report

Comments and Suggestions for Authors

Given that the manuscript is represent as a chronology of the cardiomyopaties, from the first definition to the last molecular and genetic researches, it is necessary to adapt the title to the content. The entire manuscript should be devided with subtitles into the smaller unites in relation to the hystorical chapter or thematicaly, in relation to the development of knowledge about cardiomyopathies. Thus, the hystorical overview will be organized and easier to follow, considering that there is a lot of vaulable data and information. 

Main question adressed by the research was the development of the knowledge about cardiomyopathies from the first classification and discovery to the molecular biology and genetic determination. Therefore it is suggestion for the authors that the Title of the manuscript could be „Various aspects of Cardiomyopathies; temporal review and genetic determination“
This manuscript is supplemented by a very important historical overview of cardiomyopathies including advances over time in classification, description, diganostic tools and therapy. 
Significant contribution to the methodology should be creation of the smaller chapters with sub-heading to make it easer to follow the text and have a better whole. 
The suggestions are: Introduction, Historical overview, page 2. before Cardiomyopathies Morphological and hemoynamic classification of cardiomyopathies, page 4 before Among specific... Pathophysiological classification of cardiomyopathies, before As far as inflammatory...Myocarditis, before Intracellular storage Storage diseases, page 7 before After WHO... Chronology of WHO classification, page 13 before Cardiomyopathies are usually ... Genetic background of cardiomyopathies
Conclusions consistent with the evidence and arguments presented in the manuscript. 
Reference are appropriate. 
Figures tables and images are excellent. 

Author Response

  • The title of the paper is now “Cardiomyopathies: temporal review and genetic determination”.
  • The paper has been subdivided as follow:
  • ABSTRACT
  • KEYWORDS
  • INTRODUCTION
  • HISTORICAL OVERVIEW
  • PATHOPHYSIOLOGICAL CLASSIFICATION OF CARDIOMYOPATHIES
  • STORAGE AND INTERSTITIAL CARDIOMYOPATHIES AND HEART DISEASES
  • CHRONOLOGY OF WHO CLASSIFICATION
  • STORYTELLING OF DEFINITION AND CLASSIFICATION OF CARDIOMYOPATHIES
  • GENETIC BACKGROUND OF CARDIOMYOPATHIES
  • TIME OF GENE THERAPY?
  • DNA ACTING STRATEGIES FOR GENETIC THERAPY
  • MODIFIED RNA MESSENGER
  • FINAL REFLECTIONS

Round 2

Reviewer 2 Report

Comments and Suggestions for Authors

The new title of the review better matches its content. The authors made the edits to the text of the manuscript and received permission to publish most of the Figures. It is a great pity that the authors did not consider it possible to expand the section Genetic background of cardiomyopathies, which could have significantly strengthened the manuscript.

There are a number of inaccuracies that need to be corrected.

  1. P. 13, paragraph 1 – «av» conduction - It is better to decipher or write in capital letters (AV).
  2. Figure 3. It is necessary to describe what is shown in Figures 3a and 3b and indicate the staining method.
  3. Figure 11. It is necessary to specify the description of a, b and c.
  4. There are questions about Figure 20. The caption «DCM» in Figure 20a and the caption below the Figure do not correspond to each other. Figure 20b has been modified to include a figure containing the structure of a thin filament, which was published in https://doi.org/10.1016/j.pcad.2004.07.003/. There is no indication of this and no permission to publish has been obtained. According to the description of Fig. 20 in Franz WM, Müller OJ, Katus HA. Cardiomyopathies: from genetics to the prospect of treatment. Lancet. 2001 Nov 10;358(9293):1627-37, in this figure, DCM and HCM mutations are indicated by a specific color. For clarity, an explanation should be given in the caption.
  5. Figure 22. It is necessary to add a description of c and d in the caption.
  6. Figure 24. It is necessary to indicate the source of the DNA helix images
  7. Figure 25. The PCR image is taken from the internet. It is necessary to add an indication of this.
  8. P. 18 – Karl von Rokitansky e Rudolph Virchow attended in Rome in 1894 a meeting “Morgagni and the Anatomic Concept”. – Needs to be corrected.
  9. The term ECG is usually written in capital letters. Please check in the text.

Author Response

I thank very much indeed reviewer 2 for the positive suggestions, to which I replied (see below).

Regarding the possibility to expand the section of genetic background, the actual paper is overwhelmed by pathology and I think that genetic information is sufficient. I am planning a specific paper devoted to genetics, including congenital heart diseases and other cardio-vascular diseases like atherosclerosis.

  1. 13, paragraph 1 – «av» conduction - It is better to decipher or write in capital letters (AV).

Capital letters AV replaced av

  1. Figure 3. It is necessary to describe what is shown in Figures 3a and 3b and indicate the staining method.

Figure 3a and 3b have been better described, adding haematoxylin-eosin as staining method.

  1. Figure 11. It is necessary to specify the description of a, b and c.

Figure 11 has been removed, because the illustrations are too much sophisticated. It has previously used in paper of mine (Thiene G. Storytelling of Myocardial Biopsy. Biology (Basel). 2025 Mar 18;14(3):306), but here are superfluous.

  1. There are questions about Figure 20. The caption «DCM» in Figure 20a and the caption below the Figure do not correspond to each other. Figure 20b has been modified to include a figure containing the structure of a thin filament, which was published in https://doi.org/10.1016/j.pcad.2004.07.003/. There is no indication of this and no permission to publish has been obtained. According to the description of Fig. 20 in Franz WM, Müller OJ, Katus HA. Cardiomyopathies: from genetics to the prospect of treatment. Lancet. 2001 Nov 10;358(9293):1627-37, in this figure, DCM and HCM mutations are indicated by a specific color. For clarity, an explanation should be given in the caption.

Figure previous 20 has been removed, the text is enough informative.

  1. Figure 22. It is necessary to add a description of c and d in the caption.

The caption of Figure 22 (now Figure 20) has been completed with caption also of c and d.

  1. Figure 24. It is necessary to indicate the source of the DNA helix images.

The DNA helix images employed in Figure 24 (now Figure 22) are in the public domain and do not require attribution to a specific author or permission.

  1. Figure 25. The PCR image is taken from the internet. It is necessary to add an indication of this.

Figure 25 (now Figure 23) is taken from a previous publication of the author (Thiene G. Storytelling of Myocardial Biopsy. Biology (Basel). 2025 Mar 18;14(3):306), with permission.

  1. 18 – Karl von Rokitansky e Rudolph Virchow attended in Rome in 1894 a meeting “Morgagni and the Anatomic Concept”. – Needs to be corrected.

We replaced e with and.

  1. The term ECG is usually written in capital letters. Please check in the text.

The term ECG has now been standardized throughout the text in capital letters.

Round 3

Reviewer 2 Report

Comments and Suggestions for Authors

The authors have made the necessary edits and the manuscript can be published after minor corrections.

  1. Figure 8 a and b. It seems to me that the signatures «a» and «b» are mixed up. Please check.
  2. Figure 11 – There is no Figure number.
  3. P. 14. «Long QT [Fig. 20a] and short QT [Fig. 20c], Brugada syndrome with non ischemic ST elevation [Fig. 20b]» probably needs to be replaced with «Long QT [Fig. 20a] and short QT [Fig. 20b], Brugada syndrome with non ischemic ST elevation [Fig. 20c]».

Author Response

  1. Figure 8 - We did what suggested, by anticipating the gross pathology illustration (now Fig. 8a), followed by histology (now Fig. 8b).
  2. Figure 11 - Quotation of Fig. 11 appeared already in the text (pag. 5) with figure number.
  3. Figure 20 – Accordingly to the suggestion, the sequence of Fig. 20 is now Long QT (Fig. 20a), Short QT (Fig. 20b), Brugada syndrome (Fig. 20c).